# Targeting Homologous Recombination Deficiency in Ovarian Cancer with PARP Inhibitors: Synthetic Lethal Strategies That Impact Overall Survival

**DOI:** 10.3390/cancers14194621

**Published:** 2022-09-23

**Authors:** Tao Xie, Kristie-Ann Dickson, Christine Yee, Yue Ma, Caroline E. Ford, Nikola A. Bowden, Deborah J. Marsh

**Affiliations:** 1Translational Oncology Group, School of Life Sciences, Faculty of Science, University of Technology Sydney, Sydney, NSW 2007, Australia; 2School of Clinical Medicine, Faculty of Medicine and Health, University of New South Wales, Sydney, NSW 2052, Australia; 3Centre for Drug Repurposing and Medicines Research, University of Newcastle, Newcastle, NSW 2289, Australia; 4School of Medicine and Public Health, University of Newcastle, Newcastle, NSW 2289, Australia; 5Hunter Medical Research Institute, Newcastle, NSW 2289, Australia; 6Northern Clinical School, Faculty of Medicine and Health, University of Sydney, Camperdown, NSW 2006, Australia

**Keywords:** ovarian cancer, high-grade serous ovarian cancer, PARP, homologous recombination repair, BRCA, homologous recombination deficiency, synthetic lethal, olaparib, niraparib, rucaparib

## Abstract

**Simple Summary:**

Synthetic lethality approaches to cancer therapy involves combining events to cause cancer cell death. Using this strategy, major advances have occurred in the treatment of women with ovarian cancer who have defects in the Homologous Recombination Repair (HRR) pathway. When the HRR pathway is defective, due to mutations or epigenetic changes in genes such as *BRCA1* or *BRCA2*, cells can no longer accurately repair double strand breaks (DSBs). Capitalising on this weakness, pharmacological inhibition of poly (ADP-ribose) polymerase (PARP) that function to repair single strand breaks (SSBs) leads to synthetic lethality in cells with defective HRR. PARP inhibitors (PARPis) including olaparib, niraparib and rucaparib are approved for the clinical management of women with ovarian cancer. Understanding and overcoming issues of acquired resistance to PARPis, extending these strategies to benefit more patients and combining PARPis with other drugs, including immunotherapies, are of high priority in the field today.

**Abstract:**

The advent of molecular targeted therapies has made a significant impact on survival of women with ovarian cancer who have defects in homologous recombination repair (HRR). High-grade serous ovarian cancer (HGSOC) is the most common histological subtype of ovarian cancer, with over 50% displaying defective HRR. Poly ADP ribose polymerases (PARPs) are a family of enzymes that catalyse the transfer of ADP-ribose to target proteins, functioning in fundamental cellular processes including transcription, chromatin remodelling and DNA repair. In cells with deficient HRR, PARP inhibitors (PARPis) cause synthetic lethality leading to cell death. Despite the major advances that PARPis have heralded for women with ovarian cancer, questions and challenges remain, including: can the benefits of PARPis be brought to a wider range of women with ovarian cancer; can other drugs in clinical use function in a similar way or with greater efficacy than currently clinically approved PARPis; what can we learn from long-term responders to PARPis; can PARPis sensitise ovarian cancer cells to immunotherapy; and can synthetic lethal strategies be employed more broadly to develop new therapies for women with ovarian cancer. We examine these, and other, questions with focus on improving outcomes for women with ovarian cancer.

## 1. Introduction

The application of molecular targeted therapy to treat poor outcome malignancies is revolutionising the field of cancer medicine and extending the lives of patients. Fundamental discovery science has elucidated the cellular response to DNA damage and this knowledge has been harnessed for the development of new therapies. Acknowledged as a major breakthrough, blocking poly (ADP-ribose) polymerase (PARP), a key enzyme in DNA repair, in tumours with genetic or epigenetic abrogation of proteins involved in homologous recombination repair (HRR), creates a synthetic lethal phenotype that kills cancer cells. Small molecule inhibitors of PARP are now clinically approved in many countries for the treatment of a number of malignancies including breast, ovarian and pancreatic cancers. In fact, in a relatively short time, PARP inhibitors (PARPis) have entirely altered the approach to treating a large subset of ovarian cancers. The clinical use of PARPis represents a major and impactful advance in the management of this disease. Here, we focus on women with certain types of ovarian cancer where significant extension in overall survival has been reported in response to a PARPi.

### 1.1. Ovarian Cancer and Defects in Homologous Recombination Repair (HRR)

Ovarian cancer encompasses a number of distinct malignancies that share an anatomical location, yet have different cellular origins, molecular profiles and responses to therapy [1,2,3,4,5]. High-grade serous ovarian cancer (HGSOC) is the most common histological subtype of epithelial ovarian cancer, with less common subtypes including ovarian clear cell carcinoma (OCCC), endometrioid ovarian cancer (EnOC), mucinous ovarian cancer (MOC) and low-grade serous ovarian cancer (LGSOC). An additional exceedingly rare subtype is small cell carcinoma of the ovary (SCCOHT). This review focuses on HGSOC, an aggressive malignancy, generally treated with a combination of surgery and platinum-taxane based chemotherapy. Despite this treatment regime, the majority of women relapse within two years and recurrent disease is generally viewed as incurable [6]. Five-year survival has remained less than 50% and, until very recently, options for treatment in addition to chemotherapy have been absent [7].

Molecular profiling is revolutionising the clinical management of HGSOC, with actionable targets now known. Mutation of the tumour suppressor gene *TP53* occurs in almost 100% of HGSOC [8,9]. Extensive research efforts are ongoing worldwide to target mutant p53 in ovarian and many other malignancies, for example using compounds that reactivate mutant p53 back to its wild-type form such as APR-246 (also known as PRIMA-1^MET^) [10,11,12]. To date, no mutant p53 targeting drug has been approved for routine clinical use. In sharp contrast to this are the recent successes of clinical targeting of molecular defects in the HRR pathway. Over 50% of HGSOC have a defect in this pathway due to mutations in *BRCA1*, *BRCA2*, *RAD51C*, *RAD51D, ATM* or *PALB2* [8,13,14,15,16,17], or methylation of genes including *BRCA1* [8,18,19,20,21,22] or *RAD51C* [23,24,25]. Mutations in HRR-associated genes have also been identified in OCCC and EnOC, albeit at lower frequencies than for HGSOC [26,27,28].

Collectively, tumours with a deficient HRR pathway due to genetic or epigenetic events are described as having a “BRCAness” phenotype that is frequently accompanied by higher levels of loss of heterozygosity, telomeric allelic imbalance and large-scale state transitions, due to the cell’s impaired ability to repair double strand breaks (DSBs), referred to as a genomic scar [29,30,31]. This genomic instability present as the result of BRCAness can be measured and used as a diagnostic tool for identifying HRR deficiency in tumours. By analysing these phenotypic effects of HRR deficiency, the involvement of defective HRR genes other than *BRCA1* or *BRCA2* can also be identified by implication. This includes HRR genes where their expression is determined by methylation. Commercial FDA-approved companion diagnostic (CDx) tests, FoundationOne^®^ CDx (Foundation Medicine, Cambridge, MA, USA) and myChoice^®^ CDx (Myriad, Salt Lake City, UT, USA) are now used to determine whether a woman with ovarian cancer is likely to see clinical benefit from a PARPi based on having HRR deficiency [32,33]. These tests generate a score, above which the tumour is likely HRR deficient and below which HRR proficient.

In addition to certain ovarian cancers, BRCAness is seen in other malignancies including breast [34], prostate [35], pancreatic [36], gastric [37] and colorectal cancers [38], as well as in acute leukemias [39]. While BRCAness is a clear driver of malignancy, it can be targeted using synthetic lethal strategies that involve inhibition of PARP.

### 1.2. The PARP Family and DNA Repair

The 17-member PARP family of enzymes includes PARP1, PARP2, PARP3, PARP4 (also known as Vault PARP) and tankyrases 1 and 2 (PARP5a and PARP5b) [40,41,42]. PARPs are involved in many key cellular processes including regulating transcription, translation, telomere maintenance, remodelling the chromatin landscape and, importantly in the context of this review, DNA repair [43,44]. PARPs catalyse the transfer of poly (ADP-ribose) (poly(adenosinediphosphate-ribose)) to target proteins. In this process of polyADP-ribosylation, also known as PARylation, catalytic activation of PARP synthesises poly (ADP-ribose), PAR, from its substrate nicotinamide adenine dinucleotide (NAD+) to form chains of PAR polymers. These chains attach covalently to specific amino acid residues on either PARP itself, known as auto-PARylation, or other acceptor proteins [45,46]. It is understood that PARP1 is responsible for over 90% of PARylation in the context of DNA damage, with PARPs 2, 3, 4, 5a and 5b also having PARylation activity [45,47].

During the DNA Damage Response (DDR) PARPs bind to sites of single strand breaks (SSBs) undergoing base excision repair (BER), PARylating substrates in order to facilitate recruitment of DNA repair machinery [48]. PARylation also destabilises PARP1 interaction with the SSB, uncoupling these two factors that then facilitates access for BER machinery [49]. Left unrepaired, SSBs pose a risk to genetic stability and therefore to cell survival. When the DNA replication fork encounters a SSB it can stall and collapse, causing a double strand break (DSB) that requires correction via HRR [50]. This creates a pharmacological opportunity in HRR deficient cells whereby inhibition of PARP reduces the ability of cells to repair DNA damage via the BER pathway. In this case, HRR deficient cells are unable to repair DNA damage by either HRR or BER, creating a synthetic lethal phenotype resulting in cancer cell death [51,52,53] (Figure 1). Simply stated, the combination of HRR deficiency and PARP inhibition is fatal to the cell. Further, the scaffold protein XRCC1 assembles protein complexes containing DNA polymerase β and DNA ligase III, preventing PARP1 engagement and activity during BER. This flags XRCC1 as an “anti-trapper” that may have implications for genome integrity [54].

## 2. PARP Inhibitors (PARPis)—Focus on Ovarian Cancer

PARP inhibitors (PARPis) including olaparib (Lynparza^®^; AstraZeneca Pharmaceuticals, Cambridge, UK), rucaparib (Rubraca^®^; Clovis Oncology, Inc., Boulder, CO, USA) and niraparib (Zejula^®^; GlaxoSmithKline, Brentford, Middlesex, UK) are small molecule inhibitors of PARP that have been approved by the US Food and Drug Administration (FDA), and other regulatory authorities worldwide, for women with ovarian cancer under certain conditions, including as maintenance therapy. Talazoparib (Talzenna^®^; Pfizer, Inc., Manhattan, NY, USA) is approved for treatment of advanced breast cancer and veliparib (ABT-888; AbbVie, North Chicago, IL, USA) is still being evaluated. An additional two PARPis, pamiparib (Partruvix™; BeiGene Ltd., Beijing, China) and fuzuloparib (AiRuiYi^®^, formerly fluzoparib; Jiangsu Hengrui Pharmaceuticals Co., Ltd., Lianyungang, China), have been approved in China for the treatment of women with ovarian cancer.

### 2.1. Timeline of Discovery and Clinical Adoption of PARPis

In 1963, Chambon and colleagues reported their initial discovery that nicotinamide mononucleotide enhanced the activity of a DNA dependent enzyme [55,56]. This discovery would go on to form the basis of the PARP field as we know it today. By the early 1980s, PARP had been found to play an essential role in the repair of DNA SSBs and the first PARPi was identified [57,58,59]. In 2005, the landmark discovery that BRCA dysfunction greatly sensitised cancer cells to PARP inhibition was reported [60,61]. This was proof that the concept of synthetic lethality could be adopted as a therapeutic strategy by targeting BRCA-related HRR dysfunction with a DNA repair inhibitor. For malignancies such as HGSOC where *BRCA1* and *BRCA2* mutations are prevalent [14], this discovery marked a clear turning point and new hope for molecular targeted therapy.

Almost a decade later in December 2014, olaparib became the first PARPi approved by the FDA for the treatment of advanced, recurrent ovarian cancers with germline *BRCA* mutation, or suspected germline mutation, and previous treatment of three or more lines of chemotherapy [62,63]. FDA approval of rucaparib followed in 2016, for treatment of the same indication [51,64]. Niraparib was approved by the FDA in 2017 for maintenance treatment of patients with recurrent epithelial ovarian, fallopian tube, or primary peritoneal cancer who were in complete or partial response to platinum-based chemotherapy [65]. Olaparib in 2017 [66] and rucaparib in 2018 [67] were also FDA approved as maintenance therapies under the same conditions as niraparib. While not currently approved for the treatment of ovarian cancer, in 2018 talazoparib was approved for the treatment of locally advanced or metastatic *BRCA*-mutated HER2-negative breast cancers [68]. Pamiparib was approved in China in 2021 for the treatment of germline *BRCA* mutated recurrent advanced ovarian, fallopian tube and primary peritoneal cancer in women who have had two or more lines of chemotherapy [69], as was fuzuloparib [70].

Reflecting the growing understanding that HRR-deficiency was the result of more than just *BRCA1* or *BRCA2* defects, niraparib was FDA approved for HRR-deficient advanced ovarian cancer in 2019 [71]. The combination of olaparib and the anti-angiogenic bevacizumab was FDA approved in 2020 for first-line maintenance of HRR deficient advanced epithelial ovarian, fallopian tube, or primary peritoneal cancers in complete or partial response to platinum-based chemotherapy [72]. Of significance, in 2020 the FDA-approved front-line maintenance with niraparib for platinum sensitive advanced ovarian cancer regardless of HRR status [73,74]. Other PARPis, including veliparib, are currently undergoing preclinical and clinical research and may be approved for either first-line or maintenance treatment of ovarian cancers in the future. A timeline of PARP and PARPi discovery, as well as clinical approvals is shown in Figure 2.

### 2.2. Structure and Function of PARPis

While all PARPis contain pharmacologically active nicotinamide/benzamide core structures that compete with endogenous NAD to access catalytic binding pockets of PARPs, each has a unique structure overall [75,76,77]. PARP1 and PARP2 are common targets for all PARPis; however, PARPis have different binding affinities for certain other PARP family members [78,79] (Figure 3). Antolin and colleagues summarise the affinity of olaparib, rucaparib, niraparib and talazoparib for different PARPs based on IC50 values from the literature and the ChEMBL database (www.ebi.ac.uk/chembl) [78]. PARP trapping potency also differs between PARPis. During PARP trapping, the PARP complex locks on at sites of DNA breakage, inhibiting the release of PARP and likely removing it from the process of DNA repair-associated PARylation, as well as inhibiting binding of DNA repair factors [80,81,82]. In order from highest to lowest, the PARP trapping abilities of five PARPis have been reported as talazoparib, niraparib, rucaparib, olaparib, and finally veliparib [80,81]. Pamiparib has also been reported to display PARP trapping activity [83]. It has been suggested that differences in PARP trapping activity associated with higher cytotoxicity will need to be considered when testing in combination with other cytotoxic therapies [80].

In addition to roles in DNA repair, PARPs function in other critical cellular processes. A role for PARP1 and PARP2 has been described in the maintenance of T-lymphocyte number and function [84]. PARP1 trapping has been shown to result in toxicity in healthy bone marrow [85]. PARP2 has been implicated in erythropoiesis and PARP2 deficient mice (Parp2−/−) are chronically anaemic [86]. Given these additional roles of PARP family members in important cellular processes, it is perhaps not surprising that adverse events are reported by patients taking PARPis.

## 3. Clinical Trials—PARPis and Ovarian Cancer

Data from clinical trials over the last decade inform clinical decisions made today regarding the use of PARPis for women with HGSOC [87,88,89]. Several trials have been pivotal in gaining approval from regulatory bodies worldwide. Some of these are discussed below. Trials that have focussed on treating advanced ovarian cancer, treating patients as a maintenance therapy and combining PARPis with other drugs are listed in Appendix A. While FDA approval is mainly referred to in this review, similar approvals have been granted from the European Medicines Agency (EMA), Therapeutic Goods Administration (TGA) in Australia and the National Medical Products Administration (NMPA) in China.

### 3.1. Trials Informing Clinical Use of PARPis

Study 19 (NCT00753545) investigated olaparib as a maintenance treatment in women with recurrent platinum-sensitive HGSOC who had received 2 or more lines of platinum-based chemotherapy and had a partial or complete response to their latest round. Improvement in progression free survival (PFS) was seen in women allocated olaparib versus placebo, and although not formally powered to study overall survival (OS), an advantage in OS was observed [90,91]. SOLO-1 (NCT01844986) was a landmark clinical trial that randomised women with advanced ovarian cancer based on mutations in *BRCA1* or *BRCA2* to olaparib or placebo maintenance after first-line chemotherapy. PFS in women receiving olaparib showed unprecedented improvement [92,93]. SOLO2/ENGOT-ov21 (NCT01874353) investigated women with a *BRCA* mutation and platinum-sensitive relapsed ovarian cancer, showing again, unprecedented improvement, this time in OS [94]. Olaparib is the first PARPi to be approved for use in combination therapy. The PAOLA-1 trial (NCT02477644) combined olaparib with bevacizumab to treat women with advanced ovarian cancer, observing significant improvement in PFS in patients with HRR defective tumours [95].

The outcomes of ARIEL2 (NCT01891344) saw increases in PFS in women treated with rucaparib who had platinum sensitive relapsed HGSOC with high levels of tumour loss of heterozygosity [96]. The PRIMA/ENGOT-OV26/GOG-3012 trial (NCT02655016) investigated niraparib in women with newly diagnosed advanced, platinum sensitive ovarian cancer and showed significant improvement in PFS regardless of HRR status [97]. Niraparib is the first PARPi recommended to be administered regardless of whether cells are HRR deficient or proficient. Outcomes of the clinical trial NCT03333915 has led to approval of pamiparib for recurrent advanced ovarian cancer and germline *BRCA* mutation [69,98]. Fuzuloparib has been approved for similar indications based on the outcomes of NCT03509636 and NCT03863860 clinical trials [70].

Of note, not all ovarian cancer clinical trials of new PARPis led to approval for routine use in patients. While iniparib showed early promise, it failed clinical trials in a number of malignancies including ovarian cancer, triple-negative breast cancer, squamous non–small-cell lung cancer and others, it was structurally different to other PARPis under development and was shown not to inhibit PARP at clinically relevant doses [99,100].

### 3.2. Adverse Events Associated with PARPis

The most frequent adverse events reported by patients taking a PARPi include non-haematological toxicities such as nausea, vomiting and fatigue as well as haematological side effects such as anaemia and thrombocytopenia [101,102,103,104,105]. Some side effects seem to be amplified with specific PARPis. In patients taking olaparib, nasopharyngitis and decreased appetite have been reported [63,106]. Neutropenia, insomnia, hypertension, tachycardia and palpitations are more often reported for patients taking niraparib [103,107,108]. Large dose reductions of niraparib can also be required to manage the effects of thrombocytopenia [109]. Patients taking rucaparib have reported dysgeusia, dyspepsia, greater sensitivity to the sun and other sources of ultra-violet light, itching and increased cholesterol [96,104,110]. While talazoparib has not been approved for patients with ovarian cancer, in patients with germline *BRCA* mutations and advanced breast cancer, shortness of breath has been reported [111,112]. Patients taking pamiparib [98] or fuzuloparib [70] report a decreased appetite, neutropenia and diarrhoea [98,113,114,115]. A meta-analysis has suggested that olaparib has the mildest toxicity of the PARPis, with rucaparib and niraparib reporting much higher levels of grade 3–5 adverse events [116]. Relative to the adverse events reported with platinum-based chemotherapy, PARPi-associated adverse events may, in many patients, be considered less impactful in the context of quality of life. Furthermore, some of the more common adverse effects reported in the first 6 months of PARPi treatment were able to be resolved after this time by dose interruptions or reductions, as well as supportive care [91,104,109,117].

### 3.3. Long Term Responders to PARP Inhibition

Studies are beginning to report on long term responders to PARPis. The first study to support a survival advantage for ovarian cancer patients given a PARPi was conducted by Ledermann and colleagues where 13% (18 of 136 patients) had received maintenance olaparib for 5 or more years [118]. This study reported that patients with recurrent HGSOC that was *BRCA*-mutated and platinum-sensitive who received olaparib as a sole agent for maintenance after platinum therapy achieved longer OS than patients receiving placebo, albeit not reaching statistical significance in this study. Further investigation of this, and another, cohort, classified long term responders as >2 years and short-term responders as <3 months. It was concluded that reasons for a long-term response to olaparib was likely multifactorial; however, the presence of mutations in *BRCA1* or *BRCA2* increased the likelihood of longer survival, with *BRCA2* mutations in particular being enriched amongst the long-term responders [119].

Another study investigating long term response to rucaparib classified long term response as ≥12 months, intermediate response as >6 months but <12 months, and a short response as <6 months. Of the responders in this study, 27.5% (38 of 138 patients) had a long-term response. Interestingly, patients with *BRCA* structural variants, such as deletions or rearrangements, were amongst the longer-term responders, in some instances up to 4 years. These types of variants are less likely to succumb to secondary reversion mutations, thereby removing at least one of the mechanisms of developing resistance to a PARPi [120]. In this study, high levels of *BRCA1* methylation also corresponded to longer OS. As with olaparib, extended treatment with rucaparib has been shown to be both safe and well tolerated [119,120]. With the accumulation of time, additional studies on long-term survivors of HGSOC treated with a maintenance PARPi will no doubt emerge and elucidate further predictive biomarkers of this sought after response.

## 4. Understanding and Overcoming PARPi Resistance

As with platinum-based drugs, some patients with ovarian cancer have innate resistance to a PARPi and some acquire resistance during treatment. Currently there are four mechanisms thought to influence resistance to a PARPi: (i) reactivation of HRR by secondary mutations or loss of hypermethylation, (ii) stabilisation of the DNA replication fork, (iii) reduction in the efficacy of PARP trapping and, (iv) cellular availability of the inhibitor (Figure 4).

### 4.1. Reactivation of HRR

*BRCA1/2* reversion mutations have been found in ovarian tumour tissue and cell lines from patients, with correlations or predictions made with loss of sensitivity to both platinum drugs and PARPis [13,121,122,123]. Furthermore, these mutations have been detected in circulating cell-free DNA (cfDNA) in plasma from patients both pre-treatment and after progression of their cancer, correlating with decreased responses to a PARPi [124]. Methylation of the *BRCA1* promoter that silences *BRCA1* gene expression has been reported in ovarian cancer tissue [8,20,125]. Loss of *BRCA1* gene silencing by methylation was investigated in HGSOC patient-derived xenografts challenged with rucaparib. This study showed that loss of methylation can occur post chemotherapy and that while methylation of both *BRCA1* alleles was associated with a response to the PARPi, heterozygous methylation was associated with resistance [18]. Another way cancer cells have been shown to restore HRR proficiency is by producing splice variants. An example of this is the *BRCA1-Δ11q* alternative splice isoform that lacks the majority of exon 11. The presence of a frameshift mutation in *BRCA1* was shown to result in nonsense mediated decay of full-length mRNA transcripts, but not of the *BRCA1-Δ11q* isoform. Subsequently, presence of the BRCA1-Δ11q protein isoform was linked with partial resistance to treatment with a PARPi and cisplatin [126,127,128].

Genetic and epigenetic events in other genes functioning in HRR have also been linked to loss of PARPi sensitivity. Secondary mutations in *RAD51C* and *RAD51D* have been observed to restore the open reading frame of these genes and negatively impact upon PARPi sensitivity [129]. Loss of *RAD51C* methylation has also been shown to impact upon cellular sensitivity to a PARPi [24]. Furthermore, loss of the DNA damage response factor p53-binding protein 1 (53BP1) in *BRCA1* mutant cells has been shown to restore HRR proficiency and inhibit sensitivity to PARP inhibition [130].

Aberrations of other genes including REV7 and TRIP13 can also serve to reactivate HRR. REV7 is a member of the shieldin complex that is recruited to DSBs, where, amongst other activities, it assists non-homologous end joining (NHEJ)-dependent repair of intrachromosomal breaks and sensitises BRCA1-deficient cells to PARP inhibition [131]. Loss of REV7 in BRCA1-deficient cells restores HRR leading to resistance to PARPis [132]. As a negative regulator of REV7, the TRIP13 ATPase, frequently upregulated in cancer, has also been implicated in restoring HRR, leading to resistance to PARP inhibition [133].

### 4.2. Stabilisation/Destabilisation of the DNA Replication Fork

HRR-associated proteins including BRCA1, BRCA2 and RAD51, function to stabilise stalled DNA replication forks [134,135,136,137]. They do this by protecting newly transcribed DNA at stalled or reversed replication forks from degradation by the DNA repair nuclease MRE11, thus avoiding genomic instability [138]. HRR-associated proteins that become dysfunctional due to mutation or methylation lose their ability to protect against degradation of DNA at the replication fork, therefore triggering susceptibility to PARP inhibition.

As such a critical element in the maintenance of genomic stability, it is not surprising that a number of additional factors are involved in protection of the DNA replication fork. One of these other factors is the SNF2-family DNA translocase SMARCAL1 that functions in BRCA1/2-deficient cells to stabilise the DNA replication fork [139]. Another of these factors is the MLL3/4 complex protein PTIP that functions to inhibit the recruitment of MRE11 to stalled replication forks, thus preventing their degradation and ensuring their stabilisation [140]. Inhibition of the MUS81 nuclease has also been shown to restore DNA replication fork protection in *BRCA*-mutant cells [141]. In these ways, both resistance to PARPis, as well as platinum drugs, develops based on stabilisation of the DNA replication fork, even though the original events of defective HRR signaling remain.

Given that stabilisation of the stalled DNA replication fork leads to PARPi resistance in HRR-defective tumours, strategies that encourage collapse of this fork and so overcome replication fork protection should work to overcome this resistance, thus restoring sensitivity to PARP inhibition. Acetylation of histone 4 at lysine 8 (H4K8) by the histone acetyltransferase PCAF (p300/CBP-associated) recruits the MRE11 nuclease to stalled forks, but PCAF activity has been reported as low in some BRCA2-deficient cells [142]. The RNA polymerase 1 inhibitor CX-5461 has also been demonstrated to overcome replication fork protection in patient-derived xenograft (PDX) models of HRR-deficient HGSOC involving MRE11 degradation of replication forks [143,144]. Down-regulation of the de-ubiquitylating enzyme USP1 (ubiquitin specific peptidase 1) in BRCA1-deficient cells leads to replication fork destabilisation in a synthetic lethal manner and may be another strategy to address PARPi resistance in HRR deficient tumour cells that have undergone stabilisation of the DNA replication fork [145,146].

### 4.3. PARP Trapping Efficiency

PARP trapping efficiency can be decreased by both mutations in PARP and by decreasing levels of PARP in the cell. The E3 ubiquitin ligase TRIP12 has been shown to polyubiquitinate PARP1, thus marking it for proteasomal degradation and limiting the efficacy of PARP1 trapping [147]. Furthermore, mutations in the PARP1 DNA-binding zinc-finger domains have been shown to decrease PARP1 trapping, therefore decreasing sensitivity to PARPis with a high reliance on PARP trapping to effect cytotoxicity [148].

### 4.4. Regulation of Drug Efflux Pumps

Multi-drug resistance proteins are plasma membrane pumps that actively work to transport cytotoxic agents such as chemotherapy drugs out of the cancer cell. Upregulation of these drug efflux pumps can cause drug resistance. Multi-drug Resistance Protein 1 (MDR1), also known as p-glycoprotein 1, is a transmembrane glycoprotein belonging to the superfamily of ATP-binding cassette (ABC) transporters [149]. Upregulation of this drug efflux pump has been reported as the result of promoter fusion in patients with ovarian cancer [149]. MDR1 substrates include paclitaxel [150,151,152], but there appears to be a varied response to different PARPis. There is evidence that olaparib is a MDR1 substrate, with upregulation of this drug resistance pump being a mechanism of resistance to this PARPi [153,154].

Two possible strategies to overcome or avoid PARPi resistance related to drug efflux pumps are to either reduce the expression of drug transporters or employ PARPis that do not appear to be substrates of p-glycoprotein. All-trans retinoic acid (ATRA), targets activity of the cancer stem cell marker aldehyde dehydrogenase A1 (ALDH1A) and has also been shown to down-regulate p-glycoprotein [155]. Treatment of cells with diethylaminobenzaldehyde (DEAB) also results in down-regulation of p-glycoprotein [155]. Furthermore, in a genetically engineered mouse model of *Brca2*-related breast cancer, inhibition of p-glycoprotein with tariquidar was shown to partially re-sensitise some tumours to olaparib [156]. While olaparib is a substrate of p-glycoprotein, veliparib and pamiparib do not appear to be and so resistance to these PARPis should not be mechanistically related to this plasma membrane pump [153,154]. Combination therapies that minimise the activity of drug efflux pumps, while at the same time facilitate inhibition of PARP, should be considered.

## 5. Drug Repurposing for HRR Deficient Ovarian Cancer

To date, the development of PARPis for HRR deficient ovarian cancer remains focussed on identification of new drugs. Many FDA approved drugs for cancer and non-cancer indications are pleiotropic and provide non-specific inhibition of multiple targets, including PARP, but remain under-studied for potential clinical use for HRR deficient ovarian cancer. The method of identifying new uses for approved or investigational drugs that are outside the scope of the original intended or approved medical use is termed drug repurposing [157,158]. The development of repurposed drugs that target PARP is attractive both in terms of the substantial cost and time efficiencies it offers in comparison to drug discovery [159].

The very first report of PARP inhibition was the discovery that nicotinamide has weak PARPi properties [160] and virtually all PARPis developed to date still contain the nicotinamide pharmacophore [161]. The anti-psychotic drug class, benzamides, were the second class of agents to have reported PARPi activity [162]. Analogs of both nicotinamide and benzamides have been used to extensively study the selective blocking of the NAD+ catalytic domain of PARP [163] but have not been further investigated for potential clinical use due to weak PARP inhibition. Other PARP binding domains remain understudied as potential drug binding sites including the DNA-binding zinc-finger binding domains [163,164] and the WGR domain [165].

Although the PARP inhibition reported for nicotinamide and benzamides was not at therapeutically relevant levels for HRR-deficient ovarian cancer, there remains an opportunity to screen for other drug classes that inhibit PARP. To date, there are limited examples of large-scale drug repurposing screens for PARP inhibition for HRR deficient ovarian cancer. One such screen used the Prestwick chemical library that contains 1,280 FDA-approved drugs to identify the alkylating agent chlorambucil as being toxic to *BRCA1/2* deficient cells [166]. There is potential that future repurposing screens for PARP inhibition activity of FDA approved drugs, will identify cost-effective and readily available treatment options targeted at PARP inhibition in HRR deficient ovarian cancer.

## 6. Discovering New Synthetic Lethal Relationships to Treat Ovarian Cancer

Based entirely on the concept of synthetic lethality, PARPis have clearly made a major impact on the survival of women with HRR defective ovarian cancer. Can this major advance be emulated to design new approaches for the treatment of PARPi and/or platinum-resistant tumours, as well as for ovarian, and other, cancers, that are HRR proficient? Discovery tools have expanded to meet this challenge with the advent of three-dimensional cancer models, both cell lines and organoids, that can better mimic the behaviour of cancer cells in vivo in a sustained manner, including their response to molecular targeted drugs [1,167,168,169,170,171,172]. Given that *TP53* is mutated in almost all HGSOC, the development of a synthetic lethal strategy to target the effect of these mutations would have large impact. Proteins and pathways under investigation as synthetic lethal partners to mutant p53 include the DNA damage checkpoint kinase CHK1, the nuclear kinase WEE1, ATR (ataxia telangiectasia and Rad3-related protein) and components of PI3-K (phosphoinositide 3-kinase) signaling (reviewed in [173]). Knockdown of TIGAR (TP53 induced glycolysis and apoptosis regulator) was shown to down-regulate BRCA1 and the Fanconi anemia pathway, increasing the efficacy of olaparib [174].

An increasing number of studies are focussing on achieving and/or increasing sensitivity to different PARPis, including in cancer cells that are HRR proficient. A key question being explored in these studies is can HRR deficiency in fact be induced? While many of these studies are being conducted in cancer types other than ovarian, the discoveries made have direct relevance to ovarian cancer. Using HRR proficient pancreatic cancer cells for discovery, investigators have shown that a small molecule disruptor of the BRCA2-RAD51 relationship sensitised cells to olaparib at a level comparable to *BRCA2* mutated cells [175]. Continuing with a focus on RAD51, the LRRK2 (leucine-rich repeat kinase 2) inhibitor GSK2578215A suppressed HRR, disrupting the BRCA2-RAD51 relationship in DNA repair, with cells showing increased sensitivity to olaparib [176]. Irrespective of HRR status, combinations of DNA damage response inhibitors suppressing WEE1, CHK1 and ATR have shown synergistic effects with olaparib in ID8 mouse ovarian surface epithelial cells [177].

Pre-clinical studies have also highlighted the potential of focussing on modulators of the epigenome as targets for synergistic activity with PARPis, that can induce HRR deficiency. These studies include targeting chromatin readers like bromodomain-containing proteins such as BRD4 with BET inhibitors [178,179,180] and chromatin writers such as the E3 ubiquitin ligase RNF20 [181]. Additionally, the DNA methyltransferase inhibitors 5-azacytidine and decitabine, have been shown to sensitise cancers with intact BRCA to PARPis by stimulating innate immune signaling that is in part mediated by STING (Stimulator of Interferon Genes) [182]. High throughput screening approaches using CRISPR-Cas9 and/or shRNA libraries are already identifying new synthetic lethal pairings with the potential to impact upon cancer therapy (reviewed in [183,184]).

## 7. PARPis and Immunotherapy

The remarkable benefits of immune checkpoint blockade seen for some tumours such as melanoma have not yet been recapitulated for ovarian cancer, a tumour that is largely viewed as “cold”. Cold tumours have few to no infiltrating T cells and are unable to mount an effective immune response [185]. Reasons for this include that compared to some other malignancies, ovarian tumours have a low tumour mutational burden (TMB), immunosuppressive functions of the tumour microenvironment (TME) and low expression of the targets of checkpoint blockade, including PD-L1 (programmed death ligand—1) [186,187].

Excitingly, combining a PARPi with an immunotherapy is showing early promise [188,189,190]. In preclinical models, niraparib has been shown to promote tumour immune cell infiltration by CD4^+^ and CD8^+^ T cells, induce activation of interferon signaling and work synergistically to decrease tumour volume when combined with the PD-L1 checkpoint inhibitor pembrolizumab [191]. Olaparib has also been shown to induce both intratumoral and peripheral effector CD4^+^ and CD8^+^ T cells in ovarian cancer cells lacking *Brca1* [192]. Furthermore, both niraparib and olaparib treatment has been shown to activate STING signaling that is important in mediating the proinflammatory immune response induced by DNA damage [191,192]. The addition of a STING agonist together with a PARPi in pre-clinical models of breast cancer lacking *BRCA1* increased anti-tumour immunity resulting in increased therapeutic efficacy that may also be an approach to overcoming resistance to PARP inhibition [193].

Early results of clinical trials combining niraparib and pembrolizumab show promise, including for patients with recurrent platinum-resistant ovarian cancer [194,195]. A study combining olaparib and the CHK1 (checkpoint kinase 1) inhibitor prexasertib in a small group of patients, identified partial response in some HGSOC patients with *BRCA1* mutant tumours who had demonstrated PARPi resistance [196]. Immunomodulatory effects of CHK1 inhibitors have recently been reported [197,198].

## 8. Conclusions

The synthetic lethality strategies underpinning the advent of PARPis have revolutionised the approach to clinical management of a large subset of women with HGSOC. Over the next decade, research in this field will necessarily proceed to more fully address platinum and PARPi resistant ovarian cancer. It is predicted that the field of combination therapies will expand beyond the currently approved combination of olaparib and bevacizumab, bringing the benefit of PARPis to a greater number of women with ovarian cancer. Lessons learned from targeting PARP will provide fundamental insights needed to expand synthetic lethal strategies to new combinations of genetic and/or epigenetic regulation and pharmacological inhibition.

## Figures and Tables

**Figure 1 cancers-14-04621-f001:**
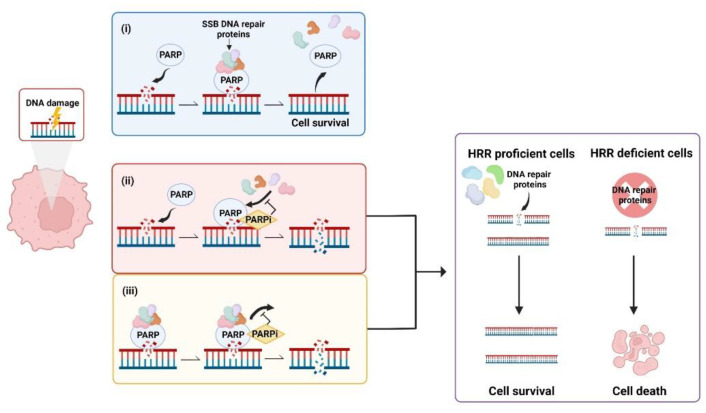
Synthetic lethality occurs when a defect in the homologous recombination repair (HRR) pathway is combined with inhibition of poly (ADP-ribose) polymerase (PARP). (**i**) PARP binds to sites of single strand breaks (SSBs), PARylates substrates and recruits DNA repair proteins. (**ii**) PARP inhibitors (PARPis) bind PARP, preventing PARylation and blocking access of PARP to DNA lesions that results in double strand breaks (DSBs). (**iii**) PARPis can also work to trap PARP at the DNA, inhibiting the dissociation of PARP from DNA and leading to the generation of DSBs. In cells with defective HRR, DSBs are unable to be repaired, leading to cell death. Created with Biorender.com.

**Figure 2 cancers-14-04621-f002:**
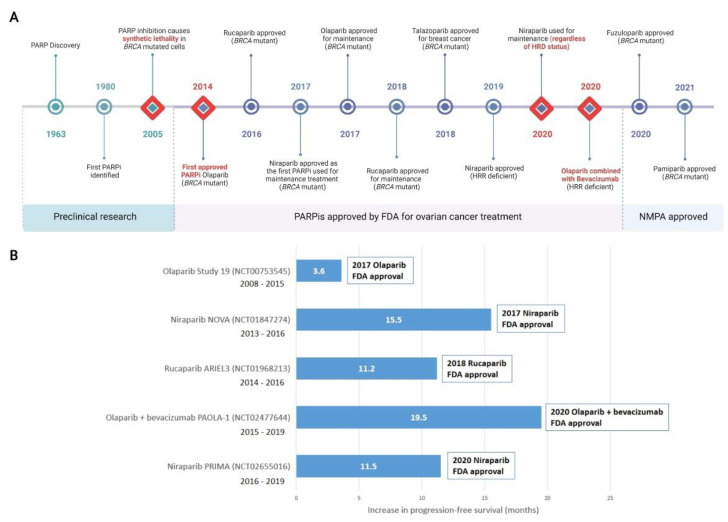
Discovery of PARP inhibitors and their introduction to the clinic. (**A**) Milestones in the discovery of PARP and clinical adoption of PARPis are recorded over time. Major milestones are indicated with a red diamond. All milestones are specific to the treatment of ovarian cancer, with the exception of talazoparib which was first approved for metastatic breast cancer and not currently approved for ovarian cancer. (**B**) Selected landmark clinical trials investigating progression-free survival (PFS) that were instrumental in clinical approval of PARPis, including in combination therapy, are shown. Increased PFS in treatment compared to placebo arms are represented. Additional months of PFS for treatment versus placebo groups are reported for the whole cohort (NCT00753545); *BRCA* mutant patients (NCT01847274 and NCT01968213); HRD positive patients including *BRCA* mutant (NCT02477644); HRD positive patients (NCT02655016). Detailed clinical trial information is reported in Appendix A. For trials noted as active but not recruiting, the primary completion date for data collection is noted. FDA, Food and Drug Administration (US); NMPA, National Medical Products Administration (China). HRD, homologous recombination deficiency; HRR, Homologous Recombination Repair; PARPi, PARP inhibitor. Created with Biorender.com.

**Figure 3 cancers-14-04621-f003:**
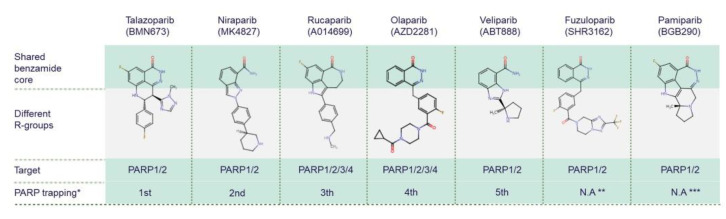
Chemical structures of clinically used PARP inhibitors. The chemical structures of PARP inhibitors (PARPis) are from ChEMBL database (www.ebi.ac.uk/chembl, accessed on 15 June 2022). * PARPis are listed by PARP trapping potency from highest to lowest [80]. ** There is no currently available data (N.A) for fuzuloparib PARP trapping potency. *** There is no currently available data for pamiparib trapping potency relative to other PARPis.

**Figure 4 cancers-14-04621-f004:**
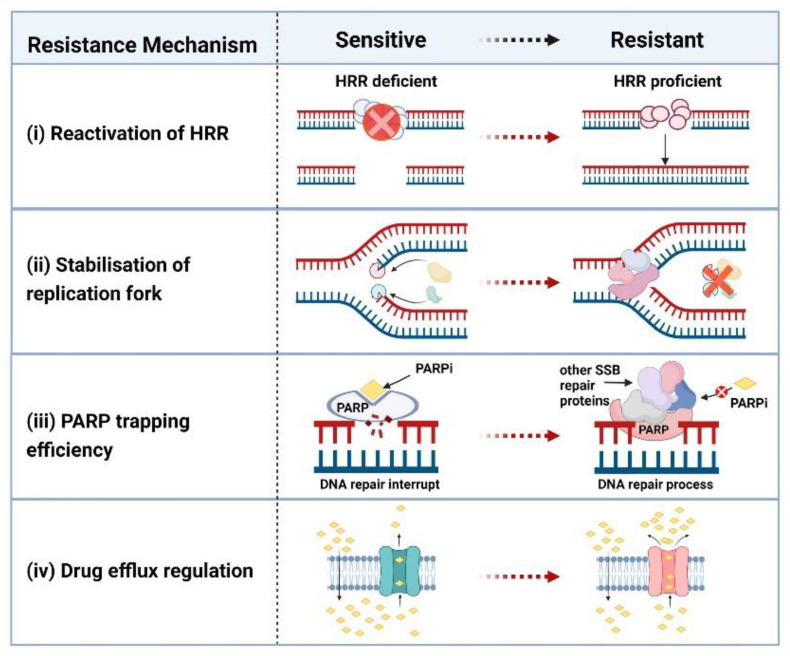
Mechanisms of PARP inhibitor resistance. There are four generally accepted mechanisms of PARPi resistance. (**i**) Homologous Recombination (HRR) restoration. The synthetic lethality is based on HRR deficiency and PARP inhibition. When HR reactivates in cancer cells for reasons including the occurrence of a reversion mutation, DNA damage can be repaired by the HRR pathway, cancer cells survive and become resistant to PARP inhibitors (PARPis). (**ii**) Stabilisation of the replication fork. Some HRR-associated proteins also function to stabilise stalled DNA replication forks. In HRR deficient cells (sensitive to PARPi), stalled replication forks will be degraded by DNA nucleases (e.g., MRE11, MUS81). PARPi resistant cells protect the replication fork by inhibiting the recruitment of DNA nucleases, therefore maintaining genomic stability. (**iii**) A PARP mutation can affect the ability of PARPi to bind PARP, leading to a decrease in PARP trapping. Mutated PARP can recruit other DNA repair proteins to correct single-strand breaks, leading to PARPi resistance and cell survival. (**iv**) Increase in drug efflux. Some PARPis (e.g., olaparib) are substrates of the transmembrane glycoprotein MDR1 (multi-drug resistance protein 1), also known as p-glycoprotein 1. By upregulating the activities of MDR1, PARPi is transported out of the cancer cells. Created with Biorender.com.

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
