# Peer review of "Targeting Homologous Recombination Deficiency in Ovarian Cancer with PARP Inhibitors: Synthetic Lethal Strategies That Impact Overall Survival"

_cancers, 2022, doi:10.3390/cancers14194621_

Round 1

Reviewer 1 Report

Minor comments
Abstract Line 30: ...molecular target therapies >> molecular targeted therapies
Lines 102/103: ...transfer of ADP-ribose (poly(adenosinediphosphate-ribose)) ...>> transfer of poly (ADP-ribose)..
Line 107: ...auto-PARYlation, or... >> auto-PARylation, or
Line 116/117: ..Cells with a deficient HRR pathway use BER as the default repair pathway to repair DSBs. This is incorrect.  Cells with a deficient HRR pathway use NHEJ (non-homologous end joining) as default pathway to repair DNA DSBs.
Line 118/119: ... inhibition of PARP removes the ability of cells to repair DNA damage via the BER pathway. >> inhibition of PARP reduces the ability of cells to repair DNA damage via the BER pathway.
Line 122: ...inhibition are fatal >> inhibition is fatal
Line 135: ...other similar agencies worldwide >> other regulatory authorities worldwide
Line 149/150: ...targeting HRR-related BRCA dysfunction.. >>  targeting BRCA-related HRR dysfunction....
Line 196: ...PARYlation >> PARylation  
Line 228: ... 2 or greater lines >> 2 or more lines
Line 305: ...four widely accepted mechanisms >> four generally accepted mechanisms
Line 425/426: ...To date, a large-scale drug repurposing screen for PARP inhibition for HRR deficient ovarian cancer has not been conducted.    >> [A report of a screen of FDA approved drugs was reported here - https://www.ncbi.nlm.nih.gov/pmc/articles/PMC6609913/ ]
Line 433: ...viewed as “cold” [This needs further explanation as to what 'cold' means, and a primary reference in support]
Line 444: ...the CHK1 (checkpoint kinase 1) inhibitor prexasertib >> [This compound is in the immunomodulator section, but it is a cell cycle inhibitor - should be elsewhere??]
Line 454: ..response to molecular target  >> response to molecular targeted

Reviewer 2 Report

This review by Xie et al focuses on leveraging PARP inhibitor synthetic lethality in homologous recombination repair deficient ovarian cancer. Overall, this review is well-written, clear, and concise, and adroitly (though briefly) summarizes the current knowledge regarding the field. As this is a well-established topic, it does not offer a novel perspective over other similar reviews that have previously been published; however, it does incrementally update existing reviews. An interesting addition might be expansion of section 1.1 to discuss research on non-BRCA HRD genes and on BRCAness detection/scoring in ovarian cancer.

The simple summary could significantly benefit from a careful rewrite for clarity and cohesion, and to better link concepts. For example, line 22-24: specifying that PARPis produce synthetic lethality in the context of HRD. Additionally, inclusion of both immunotherapy & bevacizumab here is somewhat misleading, as these are relatively minor sections of the manuscript.

Minor points:

Throughout: check consistency of abbreviations, for example HR (in figures & captions) vs. HRR (in text)

Line 30 and elsewhere: maintain consistency of ‘molecular target’ vs ‘molecularly targeted’ (line 49, line 152)

Line 60: ‘undoubtedly the most consequential advance’ – requires citation in support, or use of less conclusive modifier. Taxane-platinum combination (FDA approved 1998) remains standard of care, and so could also be described similarly.

Line 107: correct PARYlation to PARylation

Line 110-114: distinguish between BER, in which PARP1 (with XRCC1) impacts efficiency but is dispensable, and the sub-pathway of SSBR, which is initiated by PARP1. See Demin et al Mol Cell 2021 for a recent reference.

Line 116: Citation required to support the claim that BER is the primary backup pathway for HRR – HRR deficient cells also utilize classical & alternative NHEJ mechanisms to repair DSBs. Additionally, the widely accepted mechanism for PARPi synthetic lethality in HRR deficient cells is an overwhelming accumulation of unrepaired SSBs which are converted to single-ended DSBs during replication. This mechanism is articulated in the caption for figure 1, but is less clearly described in the body text.

Figure 1: Consider clearly differentiating the dual mechanisms shown in panel ii (inhibition of PARP catalytic activity and thus PARylation) and panel iii (PARP trapping with increased replication blocking effects).

Section 3.1: for completeness, the authors may consider including reference to iniparib’s biochemical/clinical failure in ovarian and other cancers

Section 4: a good overview of resistance mechanisms – is specific data available in ovarian cancer?

Section 6: the authors may consider briefly describing PARPi mechanisms of immune activation e.g. STING-mediated signaling in response to cytosolic dsDNA (Ding et al Cell Rep 2018 and others); use of STING agonists to overcome PARPi resistance (Wang et al Nat Commun 2022).

Lines 476-482: in addition to BET inhibitors, demethylating agents azacytidine & decitabine also induce PARPi synthetic lethality in OC and should be included (McLaughlin et al PNAS 2020 - disclosure: work from my lab).

Sections 6 & 7 should be reordered to place section 7 first, to maintain logical flow.

Reviewer 3 Report

In the abstract the authors state “can the benefits of PARPis be brought to a wider range of women with ovarian cancer … We examine these, and other, questions with focus on improving outcomes for women with ovarian cancer.” I personally find this question remains unanswered in the manuscript. The authors do not address the topic of patient selection and biomarkers like HRD testing for the targeted therapy of PARP inhibition. If the aim of this paper is to provide an elaborate overview, then an overview of HRD testing should be addressed as well. 

Introduction: “we focus on women with certain types of ovarian cancer where significant extension in overall survival has been reported in response to a PARPi.” In my opinion these types of ovarian cancer should be explained more in detail. In part 1.1 the authors list the histological subtypes of ovarian cancer and move forward with the statement “ This review focuses on HGSOC, an aggressive malignancy, …” The other histological types and their relation to HRD and PARPi response are not addressed. 

The authors mention molecular profiling as the revolution in management of HGSOC but do not go in dept about genomic instability scoring. The current version does not provide enough explanation about the difference between homologous recombination deficiency and pathogenic mutations of the homologous recombination repair genes. By describing BRCAness, this subject is not sufficiently explained. 

Please enhance and provide more detail in figure 1: the figure is too simple in explaining in detail the function of PARP inhibition and the DNA Repair proteins as described in section 1.2. 

Figure 2: I believe in this overview could be more detailed by adding the information about the different landmark clinical trials that led to approval of the different PARPinhibitors. Also, in the legend of this figure is the first time the term HRD is been used. 

It would give a better overview of the trials if a table can be provided, or the trials can be included in figure 2. 

3.2 adverse events associated with PARPi: the authors state neutropenia, insomnia, hypertension and tachycardia as more often reported AE in niraparib. The fact that major dose reductions are necessary with due to severe thrombocytopenia by this specific PARPinhibitor should be more highlighted as problem in clinical practice. 

The conclusion is clear and contributes well to the article. In my personal opinion this can be more elaborate, the focus of the conclusion is now on the message that more research is necessary, rather than an overall conclusion on the subject. 

Overall conclusion:  this article is well written and provides a good overview of current clinical practice of PARP inhibitors ovarian cancer. The structure and grammar of this article is very good. However, an updated version with inclusion of HRD testing is required to add novelty to the field. The current version of the manuscript does not offer enough new data to be novel and innovative between the already published overviews on PARPi in ovarian cancer. 

Reviewer 4 Report

The manuscript illustrates that inhibition of PARP activity in cancer cells is as a promising therapeutic option for ovarian cancer, predominantly HGSOC.

The manuscript is very well written and covers fundamental mechanisms of PARP inhibition in cancer cells, synthetic lethality and "BRCAness" concepts. The authors also discribe the diverse molecular mechanisms of cancer resistance to PARP inhibitors (e.g. recativation of homologous recombination, an increased activity of ABC transporters, etc.),  and the strategies to overcome this resistance.   

Despite the chapter illustrating the effectiveness of combined therapies (PARPi and immunotherapies) is mostly based on the data obtained in the preclinical models, it is novel and also looks very attractive and promising.  

The other strength of the manuscript is a broad number of completed and ongoing clinical trials, illustrating hugh efficiency of РARP inhibitors used as alone and in combination with chemo- and targeted based therapies in ovarian cancer. 

Author Response

The authors thank this reviewer for their positive assessment of our manuscript.

Reviewer 5 Report

Xie et al. provide a systematic review focusing on PARP inhibitors (PARPis)-related treatment strategies and their potential to improve outcomes for patients with ovarian cancer. It is an exciting topic and current challenge in this field. However, it is hard to understand what messages they want to deliver since they lack a connection between the title, abstract, and content. Detailed comments are as follows.

Major comments:

1. The readers would expect this review focuses on what strategies may induce homologous recombination deficiency to overcome PARPis resistance or sensitize PARPi sensitivity by reading their title "Targeting homologous recombination deficiency in ovarian cancer with PARP inhibitors: synthetic lethal strategies that impact overall survival."

However, this review only lists a few pathways involved in restoring homologous recombination repair (HRR) without describing their mechanisms. It also neglects the other BRCA1/2-independent HRR such as SHLD complex-related proteins REV7 and TRIP13 (PMID: 25799992, PMID: 31915374) or other survival pathways that are directly or indirectly involved in HR repair (PMID: 22137482, PMID: 35259479). 

Or the title should be revised if "targeting homologous recombination deficiency" is not the point the authors intended to convey.

2. The reviewer appreciates that the authors mentioned other PARPi resistance mechanisms like 4.3. PARP Trapping Efficiency and 4.4 Regulation of Drug Efflux Pumps, but these are not related to homologous recombination deficiency and should be removed.

Or they should provide relevant references that describe these two pathways that may be associated with BRCA deficiency (PMID: 25511378).

3. The abstract is well-written and raises many excellent questions in the era of PARPi treatment (lines 38-42). The readers may expect the authors would answer these questions in this review; however, some questions like "what are the key markers of a sustained response to PARPis; and can synthetic lethal strategies be employed more broadly to develop new therapies for women with ovarian cancer" did not discuss in the manuscript.

4. The authors should include the biomarkers reported in the clinical trials in Table S1-2.

Minor:

1. Table S2: PI3K/AKT pathway inhibitors are not just cell cycle inhibitors. PI3K/AKT pathway is also involved in cell proliferation, metastasis, and metabolism; it is more like an alternative survival pathway against PARP inhibitors. Therefore, please correct it as PI3K/AKT pathway inhibitors.

Round 2

Reviewer 5 Report

All questions have been addressed and the paper is well-written.